# Understanding Deep Endometriosis: From Molecular to Neuropsychiatry Dimension

**DOI:** 10.3390/ijms26020839

**Published:** 2025-01-20

**Authors:** Magdalena Pszczołowska, Kamil Walczak, Weronika Kołodziejczyk, Magdalena Kozłowska, Gracjan Kozłowski, Martyna Gachowska, Jerzy Leszek

**Affiliations:** 1Faculty of Medicine, Wrocław Medical University, 50-367 Wrocław, Poland; magdalena.pszczolowska@gmail.com (M.P.);; 2Clinic of Psychiatry, Department of Psychiatry, Medical Department, Wrocław Medical University, 50-367 Wrocław, Poland

**Keywords:** endometriosis, deep endometriosis, dementia, migraine, depression

## Abstract

Endometriosis is a widely spread disease that affects about 8% of the world’s female population. This condition may be described as a spread of endometrial tissue apart from the uterine cavity, but this process’s pathomechanism is still unsure. Apart from classic endometriosis symptoms, which are pelvic pain, infertility, and bleeding problems, there are neuropsychiatric comorbidities that are usually difficult to diagnose. In our review, we attempted to summarize some of them. Conditions like migraine, anxiety, and depression occur more often in women with endometriosis and have a significant impact on life quality and pain perception. Interestingly, 77% of endometriosis patients with depression also have anxiety. Neuroimaging gives an image of the so-called endometriosis brain, which means alternations in pain processing and cognition, self-regulation, and reward. Genetic factors, including mutations in KRAS, PTEN, and ARID1A, influence cellular proliferation, differentiation, and chromatin remodeling, potentially exacerbating lesion severity and complicating treatment. In this review, we focused on the aspects of sciatic and obturator nerve endometriosis, the emotional well-being of endometriosis-affected patients, and the potential influence of endometriosis on dementia, also focusing on prolonged diagnosis. Addressing endometriosis requires a multidisciplinary approach, encompassing molecular insights, innovative therapies, and attention to its psychological and systemic effects.

## 1. Introduction

Endometriosis is currently one of the most prominent and discussed diseases, extending beyond gynecological disorders. First described in 1860, diagnosing this condition remains a significant challenge [1]. The average diagnosis time is approximately 7 to 9 years due to its nonspecific symptoms, which are often mistaken for other conditions [2]. This issue is compounded by the insufficient knowledge about endometriosis among doctors of various specializations. Patients frequently consult physicians outside of gynecology because of symptoms affecting multiple systems.

Endometriosis does not only affect pelvic organs; it can also form grafts in areas such as the digestive system or even the brain. This review aims to investigate the neuropsychiatric comorbidities associated with deep endometriosis. Deep endometriosis significantly affects patients’ mental health, and this research focuses on examining its co-occurrence with neuropsychiatric conditions such as depression, dementia, and migraines, which significantly impair quality of life. Emerging evidence suggests a strong genetic basis for the disease, with mutations in genes such as KRAS, PTEN, and ARID1A contributing to its progression and severity. Additionally, the study seeks to explore the relationship between the severity of endometriosis and the likelihood of developing these neuropsychiatric comorbidities.

The challenges extend beyond diagnosing endometriosis to include its treatment and prevention. Ongoing research is vital, and increasing awareness among both doctors and patients could lead to shorter diagnosis times and improved patient care. By integrating molecular insights with clinical perspectives, we hope to advance the understanding and treatment of this multifaceted disease.

## 2. Methods

A literature search was conducted using PubMed, Elsevier, and ResearchGate, focusing on publications from 2004 to 2024. Search terms included combinations of “endometriosis”, “neuropsychiatric comorbidities”, “anxiety”, “depression”, “migraine”, “nerve endometriosis”, “sciatic”, “gene mutation”, and “obturator nerve endometriosis”. Only peer-reviewed studies in English, including systematic reviews, clinical trials, and case reports, were included. Pediatric and postmenopausal cases were excluded. Two independent reviewers screened articles, and discrepancies were resolved through discussion.

## 3. Endometriosis—Epidemiology and Pathophysiology

This is a disease that causes the spreading of uterine endometrial tissue outside of the normal area [3]. Those irregular locations may be the pelvic peritoneum; however, endometrial tissue may also appear in the ovaries, rectovaginal septum, or even more distant regions such as the pericardium, pleura, or the brain [1]. Endometriosis affects about 8% of the general female population; however, it is much higher in women who suffer from infertility [4].

Certain factors that increase the probability of endometriosis are family history, early age of the first menstrual period, and long and severe menstrual cycles [4]. Some research claims that there is a positive correlation between endometriosis and peripheral body fat distribution, smoking, and alcohol intake, or gastrointestinal, cardiovascular, or immunological diseases. However, those studies were limited, so further investigation in this area is needed.

The pathogenesis has not yet been entirely explained. A theory suggesting a non-uterine origin of the disease, which claims the transformation of normal peritoneal tissue to ectopic endometrial tissue, was proposed at the end of the nineteenth century [5]. Forty years later, the coelomic metaplasia theory was proposed. This hypothesis suggests that endometriosis is a kind of development from the metaplasia of the cells, which cover the visceral and abdominal peritoneum [6]. Nowadays, the most popular hypothesis is the retrograde menstruation phenomenon, which is characterized by the spread of endometrium through the fallopian tubes due to dyssynergia uterine contractions [7]. After they get to the peritoneal cavity, they may be implanted and invade further pelvic structures [5]. There are suggestions that endocrine-disrupting chemicals (EDCs) may potentially transform or stimulate the process of spreading the endometrium tissue [8]. Nevertheless, the pathogenesis of endometriosis includes the etiology of its psychological symptoms. Endometriosis can be associated with a significant psychosomatic and social burden. Cross-sectional studies have shown a higher risk of patients with endometriosis being diagnosed with depression, generalized anxiety disorder, and post-traumatic stress disorder. Previous reviews have shown that endometriosis reduces patients’ overall quality of life (QoL) and psychosocial well-being [9].

At least one-third of patients with endometriosis suffer from psychiatric disorders (mainly depression and anxiety). Patients require psychiatric or psychotherapeutic support. According to three neuroimaging studies involving patients with endometriosis, areas of the brain associated not only with pain processing but also with cognition, emotion, self-regulation, and reward probably constitute the so-called “endometriosis brain” [10].

There has also been a recent theory about human endometrial stem/progenitor cells [6]. What is sure is that the pathogenesis of endometriosis is multifactorial. There is a significant need for a precise analysis to explore the role of genetics, environmental aspects, and the immune system in patients inclining toward endometriosis development. Further research needs to be conducted to entirely explain endometriosis pathophysiology.

Research on the relationship between race and endometriosis has been limited, but emerging evidence suggests that racial and ethnic disparities exist in both the diagnosis and management of the condition. Historically, endometriosis has been perceived as predominantly affecting white women, leading to potential underdiagnosis in women of other racial backgrounds, particularly Black and Hispanic women. This disparity may be partly due to differences in healthcare access, biases in symptom recognition, and varying perceptions of pain by both patients and healthcare providers. When it comes to neuropsychiatric comorbidities like anxiety and depression, racial and ethnic factors may also play a role. Studies indicate that women from minority backgrounds often experience higher levels of stress due to systemic inequalities, which may exacerbate mental health issues. Moreover, cultural stigmas around mental health treatment in some communities may contribute to lower diagnosis rates and inadequate management of these conditions in women of color with endometriosis. More research is needed to understand how race influences both the physical and psychological aspects of endometriosis, which could lead to more equitable care [11,12].

## 4. Molecular Basis of Endometriosis

### 4.1. Gen Mutation

Endometriosis demonstrates a significant genetic basis, with mutations in specific genes, including KRAS, PTEN, and ARID1A, actively being researched to comprehend their association with the increased risk of developing DE. Understanding the molecular and genetic foundation of DE is crucial to provide therapeutic strategies and improve clinical management. KRAS (Kirsten rat sarcoma viral oncogene homolog), part of the RAS gene family, represents one of the most common oncogenes. KRAS mutations, the most common somatic mutations currently reported in endometriosis, ranging from 19.4 to 46.7%, result in increased cell proliferation and differentiation through enhanced GDP/GTP exchange and reduced GTPase activity. These mutations are frequently identified in cases of ovarian and deep endometriosis [13,14]. Moreover, a study by Orr et al. found that KRAS mutations correlate with more severe anatomical manifestations of endometriosis and thus more surgical complexity, suggesting that the mutation contributes to lesion growth, invasion, and spreading. The authors propose that KRAS mutations may serve as potential targets for non-hormonal therapeutic strategies in endometriosis [13].

The tumor suppressor gene PTEN (phosphatase and tensin homolog) is a crucial regulator of the cell cycle. Loss of heterozygosity (LOH) at the 10q23.3 locus, PTEN somatic mutations, and changes in the levels and distribution of proteins in the PTEN-PI3K/Akt signal transduction pathway are associated with endometriosis. Western blot and immunohistochemical analysis revealed decreased PTEN and increased p-Akt and p-Bad levels in ectopic endometrium of patients compared with controls (all comparisons, *p* < 0.0001). Unfortunately, research on PTEN’s role in endometriosis remains limited, and further investigations are required to understand its involvement and potential therapeutic approach [15].

Another tumor suppressor gene implicated in endometriosis is ARID1A (AT-rich interaction domain-containing protein 1A). The key component of the SWI/SNF complex is responsible for chromatin remodeling, and thus differentiation and proliferation. ARID1A mutations which occur randomly in the coding regions, the great majority being frameshift and nonsense, leading to lost expression of ARID1A, are found in preneoplastic lesions of endometrial tissue, suggesting a pivotal role in the potential transformation of endometriosis into cancer [16,17]. Furthermore, patients investigated by Yachida et al. with a mutation in ARID1A in ovarian endometriosis had a higher frequency of endometrial lesions in both ovaries [18]. Interestingly, ARID1A mutations often coexist with mutations that affect the PI3K/Akt pathway, indicating a possible cooperative mechanism between these pathways [19]. Further research into these genetic alterations could provide insight into the pathogenesis and treatment of endometriosis.

### 4.2. Epigenetic Changes

Epigenetic changes such as DNA methylation and histone modifications change gene expression without changing the DNA sequence. Increased expression levels of DNA methyltransferases (DNMT1, DNMT3a, and DNMT3b) observed in endometrial lesions alter the expression of critical genes that regulate cell growth and apoptosis. Several of endometriosis’ hallmark features such as resistance to apoptosis, invasion, and increased proliferation are linked with epigenetic changes that drive these pathological behaviors [18].

### 4.3. Hormonal Influences

Deep endometriosis (DE) is characterized by complex hormonal and molecular alterations that drive the persistence and invasiveness of ectopic lesions. A key feature is estrogen dependence, where overexpression of aromatase (CYP19A1) in DE tissues promotes excessive local estrogen production from androgens [20]. Studies suggest that individuals with endometriosis frequently show increased estrogen levels within the endometriotic tissue itself, despite normal overall estrogen levels in the body. This localized elevation in estrogen activity supports the growth and persistence of ectopic endometrial tissue, which may exacerbate the symptoms associated with the condition [21].

Progesterone resistance is another hallmark of DE, linked to reduced expression of progesterone receptors (PR-A and PR-B) or altered receptor isoform ratios. This resistance diminishes the activation of downstream pathways like HSD17B2 [22], which normally converts estradiol to its less potent form, estrone. Moreover, insufficient PR-mediated signaling impairs the suppression of pro-inflammatory and pro-fibrotic cytokines, including TNF-α, IL-6, and TGF-β, which are central to lesion fibrosis and immune evasion [21,23]. Additionally, epigenetic changes, such as hypermethylation of the PR gene promoter, exacerbate progesterone resistance, while microRNAs (e.g., miR-29c) may further modulate the expression of PRs and related pathways [24,25].

These molecular disruptions create a self-sustaining inflammatory and hormonal microenvironment, underscoring the need for therapies targeting aromatase activity, ERβ signaling, and epigenetic regulators to effectively manage DE.

### 4.4. Immune Dysregulation

Women with endometriosis have elevated levels of key pro-inflammatory cytokines, i.e., IL-1β, IL-6, and TNF-α. Moreover, IL-1β and IL-6 could be used as predictors for endometriosis. Several cytokines, including vascular endothelial growth factor (VEGF), interleukin 6 (IL-6), and tumor necrosis factor α (TNF-α), have been studied in the pathogenesis of endometriosis. Interleukin 6 is considered to play a potential role in the growth and/or maintenance of ectopic endometrial tissue [26]. Cytokines are involved in the progression of endometriosis and affect cell differentiation and proliferation. Pro-inflammatory molecules are found in the endometrial fluid. They affect the number of mature oocytes of optimal quality [2]. There is a link between reactive oxygen species (ROS) and pro-inflammatory factors that contribute to pain and lack of detoxification of lipid peroxidase products under oxidative stress. IL-6, IL-10, IL-1 beta, IL-17, and VEGF are involved in this process and increase superoxide dismutase (SOD) activity [27]. Macrophages induce anti-apoptosis of endometrial cells and release certain cytokines leading to the recruitment of more macrophages into the peritoneal cavity. Thus, excessive macrophage activity causes ectopic endometrial cells to survive and lead to the development of endometriosis lesions [28]. NK cells are suggested to play an important role in the pathogenesis of endometriosis. They take part in killing infected and malignant cells and have a role in tissue remodeling in different organs such as the uterus [29]. In endometriosis, peripheral blood NK cells (pNKs) are observed to have an increased expression of killer cell inhibitory receptors (KIRs) [29].

### 4.5. Invasion and Angiogenesis

It is suggested that Matrix metalloproteinases (MMPs) play a crucial part in the progression of endometriosis invasion. After the invasion, there is an activation of angiogenesis which is initiated by MMPs [30]. MMP-1 by stimulation of endothelial cells increases the level of VEGF receptor-2 which leads to enhanced VEGF-A-dependent signaling and, as a result, enhanced cell proliferation [31]. MPP-7 by degradation of human soluble VEGF receptor-1 enhances the bioavailability of VEGF [32].

Endometriosis-associated angiogenesis is driven, among others, by proangiogenic cytokines such as IL-1β, IL-6, IL-8, and IL-17A. Interleukin (IL)-6 in the presence of (IL)-1β stimulates angiogenesis associated with endometriosis by increasing angiogenic factors in neutrophils. IL-8 may potentiate neovascularization of ectopic implants and its presence is elevated in endometriosis patients as well as the level of vascular endothelial growth factor (VEGF) [33], which is a key mediator of angiogenesis [34].

Women with endometriosis exhibit higher VEGF levels in peritoneal fluid than women without endometriosis [35]. IL-17A further increases the production of VEGF, IL-8, IL-1β, and IL-6 [33].

### 4.6. Fibrosis Formation and Oxidative Stress

According to Somigliana et al. [36] The major component of deep endometriosis lesions is not endometrial-like tissue but fibromuscular tissue. While fibrosis is permanently present in all forms of endometriosis, it contributes to the classic endometriosis symptoms of pain and infertility [37].

Fibrotic tissue formation involves the excessive buildup of extracellular matrix components within and around inflamed or injured tissue. Tissue injury initiates the activation of platelets, leading to their aggregation and the formation of a fibrin clot. Concurrently, macrophages and myofibroblasts are activated, amplifying the inflammatory response. This cascade results in elevated levels of transforming growth factor-beta (TGF-β) and collagen deposition. Subsequently, the inflammatory response promotes the proliferation and activation of effector cells, facilitating barrier function restoration and neoangiogenesis. Deposition, remodeling, and organization of maturating scar tissue are further crowned with the recruitment of myofibroblasts [38].

Endometriotic lesions undergo repeated cycles of tissue damage and repair. This process is driven by the secretion of TGF-β, vascular endothelial growth factor (VEGF), IL-6, and IL-8 by platelets and macrophages, alongside the production of plasminogen activator inhibitor 1 (PAI-1) by endometriotic tissue.

Moreover, peritoneal oxidation protein products have also been reported to be augmented in DE lesions. The imbalance between reactive oxygen species (ROS) and antioxidants has been reported to implicate the chronic inflammatory response in the peritoneal cavity. ROS and the metalloprotease ADAM17 facilitate the release and movement of the Notch intracellular domain (NICD). The NICD subsequently enters the nucleus, where it directly participates in the transcriptional regulation of target genes, ultimately contributing to fibrotic processes [39].

Moreover, ROS contribute to local tissue destruction and enhance the formation of adhesions associated with ectopic endometrial cells. According to Scutiero et al., ROS markers (hydrogen peroxide, peroxidase, catalase, c-Fos, and c-Jun, 8-OHdG, MDA) identified in endometriosis cells are linked to DNA damage, inflammation, and cellular dysfunction and play a significant role in the control of endometrial cell proliferation. Their description is to be found in Table 1 [40].

## 5. Endometriosis and the Difficulties in Making a Diagnosis

Endometriosis, a condition affecting around 10% of women of reproductive age, is marked by the presence of tissue resembling the endometrium outside the uterus. This leads to both systemic and localized inflammation, resulting in symptoms such as chronic pelvic pain, painful intercourse, infertility, and dysmenorrhea. These physical symptoms are frequently accompanied by profound mental health challenges, including heightened levels of anxiety and depression, which are intensified by the persistent nature of the condition and the associated pain. Research by Missemer et al. [41] revealed that 45.8% of young women with endometriosis experienced moderate levels of anxiety, while 33.4% reported moderate depression. In parallel, da Silva et al. et al. [42] noted that individuals with chronic pain related to endometriosis exhibited higher rates of depression compared to those without such pain. These findings underscore the importance of comprehensive care that integrates both physical and mental health support for individuals living with endometriosis [41,42].

Endometriosis presents with diverse phenotypes, making it challenging to classify and diagnose. Symptoms vary widely and often do not correlate with the type or location of lesions. This heterogeneity contributes to diagnostic delays, as signs such as pelvic pain or infertility overlap with other conditions. While imaging techniques like ultrasound and MRI expedite the detection of some lesions, the development of a reliable non-invasive diagnostic test has been slow. A precise classification system for endometriosis is urgently needed to stratify patients into clinically relevant subgroups, facilitating personalized treatment strategies [43].

The approach to treating endometriosis is tailored based on factors such as the severity of symptoms, the location of the disease, the patient’s age, and fertility considerations. Hormonal therapies are typically the first-line option, with combined oral contraceptives and progestins promoting a hyper-progestogenic state to inhibit ovulation and reduce the size of endometriotic lesions. Despite this, nearly one-third of patients experience resistance to progesterone, which may be associated with chronic inflammation, alterations in gene expression, or exposure to environmental toxins. When first-line treatments are insufficient, second-line therapies like GnRH agonists are used to suppress ovarian steroid production, though they can cause side effects such as mood disturbances, hot flashes, and an elevated risk of osteopenia. Third-line treatments, including Danazol and gestrinone, work by lowering estrogen levels through androgenic mechanisms but are limited by side effects such as weight gain, oily skin, and excessive hair growth [44,45,46].

Surgical management, with laparoscopic excision as the preferred method, is typically reserved for severe cases or situations where medical treatments have proven ineffective. Laparoscopy is favored over laparotomy due to its shorter recovery period and more favorable cosmetic outcomes. Nonetheless, recurrence of symptoms remains a major issue, with more than half of patients requiring repeat surgical interventions within five years. Complementary approaches, including acupuncture and electrotherapy, may help alleviate pain but do not target the root cause of the disease. Advances in imaging technologies have enhanced the preoperative evaluation of endometriosis, particularly for identifying deep infiltrating lesions. However, accurately stratifying these lesions based on size, location, and depth continues to present challenges [47,48,49].

Endometriosis profoundly impacts quality of life, shaping critical decisions regarding education, career, and personal relationships. Chronic pain and dyspareunia play significant roles in diminishing productivity and straining social and intimate connections. Although various treatment options exist, the neuropsychiatric consequences of the condition often receive insufficient attention. Future studies should focus on holistic treatment strategies and advancing non-invasive diagnostic tools to enhance the well-being and outcomes of women living with endometriosis [41,45].

## 6. Deep Endometriosis

### 6.1. Definition, Pathology, and Comparison with Superficial Endometriosis

Deep endometriosis (DE) is a subtype of endometriosis, a chronic, often progressive, inflammatory disease characterized by the extra-uterine growth of endometrial tissue. Endometriosis affects up to 10% of women of reproductive age, with rare occurrences in postmenopausal women. Approximately 1% of women present a severe form of the disease [50,51]. DE is associated with pelvic pain, infertility, and bleeding problems. Various factors, including pollution (dioxin, PCB, radioactivity), food intake (alcohol, caffeine), lifestyle, postponement of first pregnancy, chemical disruptors, and stress, have been suggested to influence its development. Some theories propose immune dysfunction and abnormal differentiation of tissues as contributing to the disorder’s development [49]. DE has undergone definitional updates over the years. Initially defined as “adenomyoma—the endometriosis infiltrating the peritoneum by >5 mm”, it is now understood as “endometrial stroma and glands in fibromuscular tissue (adenomyosis externa)”, which does not require a specific depth measurement and leads to an estrogen-dependent chronic inflammatory response [51,52,53].

Most DE lesions exceed 1 cm in diameter and are typically singular, with rare instances of multiple lesions (two, and even less commonly three). The development of these lesions can be linked to a benign tumor, originating from the pouch of Douglas and potentially extending through the pelvis to the uterine artery, vagina, uterosacral ligaments, bladder, uterus, diaphragm, nerves, and muscles of the bowel wall [52]. Notably, endometrial lesions are preferentially located in the bowel walls along the nerves in the muscular layer. Spreading across anatomical compartments, these lesions cause dysfunction and impairment of the quality of life, manifesting as severe pelvic pain, lower abdominal pain, back pain, dysmenorrhea, dyspareunia, fatigue, headaches, and gastrointestinal problems. These symptoms often lead to mood swings, disturbance in social life, and lowered self-esteem [51,54].

To sum up, superficial and deep endometriosis differ significantly in their location, extent, and clinical implications. Superficial endometriosis primarily affects the peritoneum, the thin membrane lining the abdominal cavity, and involves lesions that are confined to this surface layer. These lesions are typically less invasive and may cause localized inflammation and pain. In contrast, deep infiltrating endometriosis (DIE) extends beyond the peritoneum, invading tissues and organs such as the bowel, bladder, or uterosacral ligaments. DIE is often associated with more severe symptoms, including chronic pelvic pain, organ dysfunction, and infertility, due to its deeper penetration and the involvement of nerve-rich areas. While both types can significantly affect a patient’s quality of life, the invasive nature of DIE often necessitates more complex diagnostic and treatment strategies compared to superficial endometriosis [55,56].

### 6.2. Correlation Between Deep Endometriosis and Migraine

Migraine is one of the most common neurological problems in primary care [57]. It is more frequent in women than men and often affects patients at a young age. Migraine is the first reason for disability in young women [58]. The definition of migraine is a mild, recurrent syndrome of headaches associated with other symptoms of nervous system dysfunction in various combinations. This condition can often be diagnosed by the factors that activate it, known as triggers [59].

In recent research, there has been a correlation between migraine and endometriosis discovered. A study conducted by Selntigia et al., has proved that severe adenomyosis and posterior and anterior deep infiltrating endometriosis are more frequent in women with migraine [60]. Also, the research of Maitrot-Mantelet et al. showed that in women with endometriosis, migraine is more frequent [61]. The highest risk in migrainous women was for the occurrence of ovarian endometrioma and deep infiltrating endometriosis [61].

Patients suffering from both migraine and endometriosis experienced notably greater pain intensity, a higher number of migraine days each month, and elevated Headache Impact Test (HIT-6) scores compared to those without endometriosis. Additionally, dysmenorrhea emerged as the most frequently reported symptom among women with both conditions. Those afflicted by both endometriosis and migraine also exhibited higher Visual Analog Scale (VAS) scores for all common endometriosis symptoms [60].

### 6.3. Correlation of Deep Endometriosis, Depression, and Anxiety

The main psychological problems which affect women suffering from endometriosis are anxiety and depression. Robert et al. indicated that 29% of women had anxiety and about 14.5% depression. [62]. One of the problems which affects women suffering from endometriosis is depression. This may be the effect of the pain. Ribiero et al. have observed that among 40 women who suffered from deep endometriosis, 77.1% of the patients had anxiety and depression [63]. Chronic pelvic pain (CPP), one of the earliest symptoms of endometriosis, can lead to physical, psychological, and social repercussions for patients, as the condition restricts and modifies their daily activities [64]. Interestingly, 77% of patients in Ribiero’s study exhibited anxiety and depression simultaneously [63].

Another factor contributing to depressed mood in women with endometriosis could be sexual impairment. The quality of sexual life significantly influences the overall quality of life. Approximately two-thirds of women with endometriosis experience sexual dysfunction, which extends beyond just deep dyspareunia [65]. Dyspareunia, or painful intercourse, is a common issue experienced by women with endometriosis, affecting an estimated 32% to 70% of those affected by the condition [66]. Nevertheless, pain during penetration is not the sole cause of sexual dysfunction. Endometriosis can profoundly affect social interactions and intimate relationships [67].

Anxiety may stem from the emotional toll of subfertility, the potential recurrence of the disease, and the uncertainty surrounding repeated surgeries and prolonged medical treatments. [68]. Moreover, anxiety and depression can heighten pain perception, while the presence of pain can further deteriorate the psychological state in such situations [69].

Endometriosis may be linked to a higher risk of bipolar disorder; however, relevant data are limited. The incidence rate of bipolar disorder is 1.04 per 1000 person-years in comparison with healthy women—the incidence rate is 0.56 [70]. The association between bipolar disorder and endometriosis may be linked to chronic pelvic pain and quality of life deterioration. It is necessary to conduct more research on whether bipolar disorder is an effect of endometriosis or a concurrent state [71].

### 6.4. Deep Endometriosis, Neurodegenerative Diseases, and Mental Health

Endometriosis symptoms can adversely affect patients’ psychological well-being and significantly impair their mental health. Currently, it is not definitively known how endometriosis affects the risk of dementia. Elevated estrogen levels contribute to the development of endometriosis, while reduced estrogen exposure is linked to neurodegenerative diseases such as Alzheimer’s and Parkinson’s disease [72]. Opposite conclusions came from a case–control study of Danish women with endometriosis. Latourelle et al. investigated whether endometriosis has a protective effect against Parkinson’s disease and found that patients with endometriosis showed a moderately increased risk of Parkinson’s disease [73]. Furthermore, oral contraceptives and hysterectomy, commonly used in the treatment of endometriosis, may also increase the risk of Parkinson’s disease [74]. In contrast, some studies have suggested a lower risk of dementia, including Alzheimer’s disease, associated with oral contraceptive use, although further research is needed to confirm these findings [75]. According to three neuroimaging studies of patients with endometriosis, regions of the brain involved in pain processing, cognition, self-regulation, and reward are likely part of what is termed the “endometriosis brain”. It remains unclear whether the neurobiological changes observed in these patients are due to chronic pain, comorbid psychiatric conditions, or the endometriosis itself. More research is necessary to explore the co-occurrence of endometriosis and psychiatric disorders, including neurodegenerative diseases, given the limited high-quality data on psychiatric comorbidities and neurobiological correlates in endometriosis [10]. Patients who have endured long-term endometriosis tend to experience heightened levels of perceived stress, indicating that the chronic nature of the disease independently influences stress perception [76]. Limited research suggests that women diagnosed with endometriosis may encounter a higher likelihood of experiencing psychosocial disorders or psychological distress. Nevertheless, the precise relationship between these conditions and endometriosis remains uncertain, leaving open questions about whether these disorders arise directly from the disease or are connected to the chronic pain and inflammation characteristic of gynecological conditions like endometriosis [71]. Presently, the optimal approach to patient care involves directing patients to specialized centers equipped with multidisciplinary teams comprising gynecologists, colorectal surgeons, and urologists proficient in advanced laparoscopic techniques. These centers are recognized as the gold standard due to their comprehensive capabilities in managing complex cases of endometriosis. By bringing together diverse expertise, they ensure that patients receive integrated and tailored treatment strategies that encompass both surgical and non-surgical interventions. This collaborative approach not only enhances diagnostic accuracy but also improves the overall quality of care, addressing the multifaceted aspects of the disease and optimizing patient outcomes [77].

## 7. Sciatic and Obturator Nerve Endometriosis

Endometriosis involving peripheral nerves is uncommon, with the sciatic nerve being the most frequently affected. Approximately 34% of cases of nerve-related endometriosis occur without associated peritoneal changes [78]. The sciatic nerve, the largest nerve in the human body, originates from the L4 to S3 nerve roots, exits the pelvis below the piriformis muscle, and innervates the posterior thigh and lower leg compartments [79]. Sciatic nerve endometriosis often presents as “catamenial sciatica”, characterized by worsening sciatica symptoms during menstruation [80]. While deep infiltrating endometriosis (DIE) involving major pelvic nerves or sacral roots is exceedingly rare, affecting less than 0.1% of cases, its clinical and diagnostic challenges are significant, as information about this condition is limited primarily to case studies.

Untreated sciatic nerve endometriosis may result in permanent nerve damage due to nerve fibrosis, a process mediated by repetitive tissue injury and healing cycles. Mechanisms include epithelial–mesenchymal transition (EMT) and fibroblast-to-myofibroblast transdifferentiation (FMT), driven by factors such as Tumor Growth Factor-β1 and neuropeptides, culminating in increased collagen deposition and cellular contractility [74]. This is presented in Figure 1.

Bindra et al. described a 29-year-old woman presenting with cyclical pain and mobility issues in her right leg. Initial symptoms, including dysmenorrhea and buttock pain radiating to her leg and foot, began three years after laparoscopic removal of an endometriotic cyst. Imaging revealed a 4 cm endometriotic lesion involving the sciatic nerve, which was surgically excised. Post-surgery, the patient’s cyclical pain and mobility issues were resolved completely [80]. Similarly, Zamurovic et al. reported a 45-year-old patient with chronic leg pain and tingling. MRI revealed gluteal muscle atrophy and an ill-defined mass near the sciatic nerve. Subsequent neurosurgical intervention confirmed calcified fibrous endometriosis. While pharmacological therapy with GnRH analogs provided temporary remission, surgical treatment remains the most definitive approach [74].

Passover’s study of 452 women with sciatic nerve endometriosis highlighted the aggressive nature of this condition. Among patients with symptoms lasting one to three years, 30% exhibited neurological deficits, and 80% had peritoneal fibrosis. Beyond three years, most patients showed severe nerve damage. These findings underscore the importance of early diagnosis in women with recurrent pelvic pain, as delayed treatment can result in irreversible nerve damage. Sciatic nerve endometriosis likely originates from undifferentiated cells within the nerve itself [74].

Obturator nerve endometriosis is an even rarer manifestation, with only eight documented cases, accounting for about 1% of peripheral nerve involvement [81]. The obturator nerve, running along the inferolateral bladder, facilitates leg adduction through muscles such as the adductor longus, brevis, and magnus. Symptoms include thigh pain, weakness, and difficulty with leg adduction. Magnetic resonance imaging (MRI) is the preferred diagnostic tool, revealing nodules with variable signal intensities characteristic of endometriosis. Treatment involves radical laparoscopic excision, which requires precise dissection to expose the obturator nerve and remove the affected tissue while preserving surrounding structures [82].

The aggressive progression and debilitating effects of nerve-related endometriosis demand timely and multidisciplinary intervention. Early recognition and advanced imaging are crucial for preventing severe neurological sequelae and improving patient outcomes. These cases highlight the need for increased awareness and specialized expertise in managing this rare yet impactful manifestation of endometriosis.

## 8. Conclusions

The impact of deep endometriosis on neuropsychiatric conditions remains an area of ongoing research. With the increasing prevalence of endometriosis, interest in its pathophysiology is also rising. Diagnosis of this condition remains challenging for specialists, as symptoms are often nonspecific and may extend beyond the genitourinary system. Most people associate DE with pelvic pain, abnormal uterine bleeding, and infertility. However, the correlation between endometriosis and conditions such as migraines, anxiety, depression, and even dementia should not be neglected. Research indicates that patients with endometriosis who experience migraines report higher pain intensity and more frequent migraine days, leading to elevated HIT-6 scores. Chronic pelvic pain, one of the earliest symptoms of deep endometriosis, results in a greater risk of anxiety and depression. Physical pain leads to psychological and social disturbances, limiting daily activities and reducing patients’ quality of life. Sexual dysfunction significantly affects mental health, with factors like dyspareunia, bladder pain, dysmenorrhea, and unemployment affecting women’s sexual quality of life [83]. Main outcome from this article is presented in the Table 2.

The fear of infertility and uncertainty about future relationships due to, for instance, prolonged medical treatment contribute to increased anxiety. Regarding the mutual influence of high estrogen levels on endometriosis as well as Alzheimer’s or Parkinson’s disease occurrence, patients suffering from DE may be at greater risk of neurodegenerative disorders. Additionally, endometriosis can affect peripheral nerves, most commonly the sciatic nerve, and very rarely the obturator nerve. This can worsen sciatica during menstruation, causing pain radiating from the buttock to the leg, along with mobility limitations. It is crucial to speed up the diagnostic process for endometriosis. Nowadays, laparoscopy remains the gold standard, following comprehensive clinical evaluations. Furthermore, it is important to raise awareness of endometriosis beyond gynecology, involving other specialties due to its wide range of symptoms.

Future research should focus on understanding the pathomechanisms behind neuropsychiatric comorbidities, expanding neuroimaging studies on altered pain processing, and developing precision medicine approaches. Longitudinal studies on psychological outcomes, improved diagnostic techniques for nerve endometriosis, and exploring non-invasive therapies like neuromodulation could improve patient care and quality of life.

## Figures and Tables

**Figure 1 ijms-26-00839-f001:**
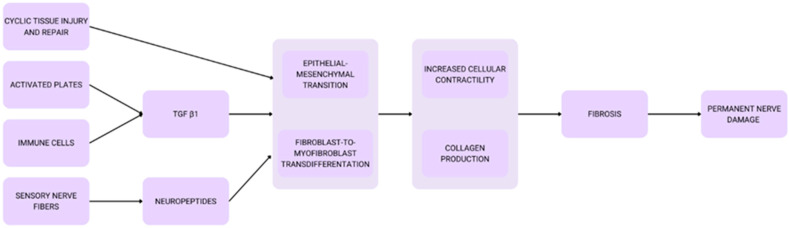
Pathophysiology of nerve impairment.

**Table 1 ijms-26-00839-t001:** Factors influencing endometriosis modulation.

ROS Marker in Endometrial Tissue	Description	Reference
hydrogen peroxide	Superoxide anion, with higher concentration in endometriotic cells than in endometrial cells	[40]
glutathione peroxidase	Antioxidant enzyme, with higher concentration in endometriotic cells than in endometrial cells	[40]
catalase	The antioxidant enzyme has a lower concentration in endometriosis cells than endometrial cells	[40]
c-Fos and c-Jun	Members of mitogen-activated protein (MAP) kinase/extracellular signal-regulated kinase (ERK) pathway	[40]
8-Hydroxy-2′-deoxyguanosine (8-OHdG)	Marker of oxidative stress damage, with higher concentration in endometriotic cells than in endometrial cells	[40]
malondialdehyde (MDA)	A byproduct of lipid peroxidation	[40]

**Table 2 ijms-26-00839-t002:** Summary of neuropsychiatric and peripheral nerve associations with endometriosis.

Disease	Association with Endometriosis		Pain Perception
Migraine	Migraine is more frequent in women with endometriosis, with the highest risk in ovarian endometrioma and deep infiltrating endometriosis.		Patients with both migraine and endometriosis reported higher pain intensity
Depression	Overall, 77.1% of the patients suffering from endometriosis had anxiety and depression, from which 77% had anxiety and depression simultaneously.	Endometriosis can have a significant effect on social life as well as on sexual life. Dyspareunia, or painful intercourse, is reported by 32% to 70% of women who have endometriosis.	Anxiety and depression increase pain perception
Anxiety		Anxiety in the context of endometriosis encompasses the emotional toll of subfertility, the potential for disease recurrence, and uncertainty regarding future outcomes.	
Bipolar Disorder	Endometriosis may be linked to a higher risk of bipolar disorder.		Chronic pelvic pain and quality of life deterioration
Dementia	Elevated levels of estrogen contribute to the development of endometriosis, whereas reduced estrogen exposure is associated with neurodegenerative diseases such as Alzheimer’s disease.		Alterations in pain processing
Sciatic Nerve Endometriosis	The occurrence of deep infiltrating endometriosis (DIE) involving sacral nerve roots or major pelvic nerves is less than 0.1%.	Sciatic nerve endometriosis is associated with pain in the buttock, radiation to the leg and heel, and mobility disorder. Surgical treatment is the most common.	Pain typically occurs as “catamenial sciatica” which means worsening of sciatica during menstruation
Obturator Nerve Endometriosis		Endometriosis of the obturator nerve is only about 1% of peripheral nerves associated with endometriosis. It is associated with thigh pain, weakness, and impaired adduction of the legs. The treatment is radical excision of the nerve using laparoscopy.	-

## Data Availability

All data are available from the authors upon request.

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
