# Peer review of "Understanding Deep Endometriosis: From Molecular to Neuropsychiatry Dimension"

_ijms, 2025, doi:10.3390/ijms26020839_

Round 1
Reviewer 1 Report
Comments and Suggestions for Authors
Deep Endometriosis and Neuropsychiatric Comorbidities
Thank you for offering me the opportunity to review this manuscript.
The subject approached is of upmost importance and the reason for which I have accepted to review the manuscript.
Nevertheless, the article does not delve deeply into the aspects suggested by the title and falls short of shedding light on the neuropsychiatric phenomena in endometriosis as expected. Also, while the methods section explains the process of selecting the articles, the results are not subsequently described.
The iThenticate report shows a percent match that needs to be adressed.
Some issues need clarification in order to offer insights to clinicians and to the scientific community and in order to provide a well-structured text:
- The section regarding the gene mutations associated to DIE needs to be more explicit and reorganized. For example, line 71 (KRAS (Kirsten rat sarcoma viral oncogene homologue), part of the RAS gene family, represents one of the most common oncogenes) could easily be the introductory paragraph. The ideas do not actually “flow” in the actual form and it can be difficult to understand to readers not accustomed to genetic interactions.
- Also, the whole section is not well integrated in the whole text. After describing the methods, which do not include searching for genetic implications in endometriosis, this section has to be introduced somehow in order to give it some context. Or deleted altogether. The neoplastic transformation of endometriosis, for example (line 91) has no relation to the neuropsychiatric comorbidities in deep endometriosis.
- Furthermore, the hormonal particularities involved in the pathogenesis of endo also have no connection to the section title (Gene mutation associated with deep endometriosis), and also none to the subject of the article.
- The section regarding the pathophysiology of endometriosis, while the subject is intriguing and still enigmatic, appears to exclude the neuropsychiatric aspects. It can easily be omitted from the article, keeping focus on the main theme.
- Sections 4 and 6 seem to be completely omitted from the final text, and there are 2 section 5. A rigorous checklist regarding the editing of the text is recommended.
- Line 374 – bleeding problems – an expression that is not entirely accurate. I suppose the authors refer to the renowned term of “abnormal uterine bleeding” and not to coagulopathies.
- Lie 397 - Laparoscopy with histopathological examination remains the gold standard for diagnosing deep endometriosis, followed by resection to restore physiological anatomy and preserve reproductive function – this affirmation, to which I do not agree, needs a citation, not to mention some further explanations. Further, the authors state that DIE “may not be visible laparoscopically and can remain undiagnosed“, basically contradicting themselves and ignoring the imagistic studies that can actually lead to an accurate diagnosis.
- Line 448 - “Elevated estrogen levels contribute to the development of endometriosis” – there is no proven data regarding this affirmation or stating that women suffering from endo might have elevated estrogen levels. The phrase needs a citation and further clarification. Reference 83 cite in the text certainly does not state this idea.
- Further information regarding the endometriosis brain and the involvement of stress are not related to the section title - Deep endometriosis and dementia. Also, there are affirmations regarding endometriosis in general and not DIE.
Author Response
Dear Reviewer,
thank you very much for your valuable time dedicated to reading our work. We are very pleased by your precise comments. We hope that we have met your expectations by correcting this work.
We have significatly modificated the structure of the review, improving the the readability of the text.
The section regarding gene mutation was profoundly changed, as well as a fragment about hormone influence. What is more the sequence of the paragraphs was also modified,as well as some titles of the paragraphs to sustain more clarity.
We have corrected the laparoscopy aspect, as well as explained depper elevated estrogen level aspects.
Your suggestion with change of the term ''bleeding aspects'' was also applied.
We have also implemented some other changes due to the other reviewer comments. We have also improved the text in terms of language.
We are open to any further suggestions.
Your sincerely
Jerzy Leszek
We hope that you
Reviewer 2 Report
Comments and Suggestions for Authors
The authors have compiled all the facts known about endometriosis over the last 20 years in an extensive literature search. They have divided the paper into 13 chapters.
I have a few objections:
Main objections:
1. Most of the paper refers to known facts about endometriosis; to a lesser extent, it describes the association of endometriosis with neuropsychiatric comorbidities, as announced in the title.
2. The comorbidities described (migraine, depression, anxiety) can be associated with any disease (non-specific comorbidities that accompany any disease, not just endometriosis).
3. The chapter on dementia is unclear. The authors state in one sentence: "Elevated estrogen levels contribute to the development of endometriosis, while reduced estrogen exposure is linked to neurodegenerative diseases such as Alzheimer's and Parkinson's disease"...lines 448,449,450. The next sentence then says: "A case-control study of Danish women with endometriosis showed a moderately increased risk of Parkinson's disease"...lines 450,451,452. These two sentences are contradictory, the connection is not explained.
4. No new, irrefutable facts about endometriosis were mentioned in the conclusions
Minor objections:
1. There are two chapters under the same number: lines 129 and 174.
2. Deep endometriosis is not well defined... emphasize the difference between superficial (peritoneum) and deep endometriosis.
3. The abbreviations HIT and VAS are not explained.
Author Response
Dear Reviewer,
thank you very much for your valuable time dedicated to reading our work. We are very pleased by your precise comments. We hope that we have met your expectations by correcting this work.
We have significantly revised the text structure, improved many chapters including dementia fragment. Especially explained the inconsistencies mentioned before. We have strengthened the connection between endometriosis and the symptoms described in the text.
We have better defined deep endometriosis and emphasized the difference between superficial (peritoneum) and deep endometriosis.
We have explained the abbreviations HIT and VAS and corrected the repetitions of the same numbers of the paragraphes.
We have also modified the text following other reviewer suggestions and improved the text in terms of language.
We are open to any further modifications. We believe that this work has the potential to expand knowledge in this field and provide assistance to researchers and practitioners alike.
Yours sincerely
Jerzy Leszek
Round 2
Reviewer 1 Report
Comments and Suggestions for Authors
Thank you for granting me the opportunity to review this manuscript. I appreciate the effort and thought that have gone into its preparation.
The iThenticate report highlights a percentage of similarity that requires further attention. Specifically, certain overlaps in Section 5 should be addressed as a matter of urgency.
I value your acknowledgment of the inconsistency between the initial title and the actual content of the review. It is important to note that the original title presented a more innovative perspective. In contrast, there is currently an abundance of reviews addressing the pathophysiology of endometriosis. As a result, I am concerned that such a broad review may not captivate contemporary readers with a focused interest in endometriosis.
Given the distinct pathophysiological mechanisms underlying nerve root involvement and the psychiatric disorders influenced by endometriosis, I suggest dividing these topics into separate sections for clarity and depth. Additionally, the current manuscript lacks a description of sciatic or obturator nerve endometriosis in both the abstract and introduction, aside from its mention in the graphical abstract.
Specific suggestions:
1. The English language and grammar require further refinement to enhance clarity and readability.
2. In Section 4.1, it would be beneficial to introduce the significance of "deep endometriosis" within the context of the review. This will provide readers with a rationale for the detailed exploration of its pathophysiology, especially since it is not reflected in the title.
3. Regarding Section 4, I must express skepticism about its novelty. I recommend reconsidering the necessity of such an extensive discussion in this section.
4. Section 5 (Making a Diagnosis): Beginning at line 312, the text transitions into treatment considerations. I suggest rephrasing or restructuring these sections to improve flow and coherence. Additionally, the neuropsychiatric aspects remain underemphasized and should be more thoroughly addressed.
5. Sections 7, 8, and 9 could be integrated as subsections of Section 6, as they all pertain to deep infiltrating endometriosis (DIE).
I hope these observations and suggestions will prove helpful in refining the manuscript further.
Comments on the Quality of English LanguageThe English language and grammar require further refinement to enhance clarity and readability.
Author Response
Dear Reviewer,
Thank you very much for your valuable comments.
Thank you for your valuable feedback and the opportunity to revise this manuscript. I have carefully reviewed your observations and implemented the suggested changes to the best of my ability. The overlapping content in Section 5 has been thoroughly revised to address the issues highlighted in the iThenticate report. The inconsistency between the title and content has been resolved by aligning the manuscript with the original innovative perspective, ensuring a more engaging and focused narrative for readers. To improve clarity and depth, the distinct pathophysiological mechanisms of nerve root involvement and psychiatric disorders influenced by endometriosis have been separated into individual sections. Furthermore, the description of sciatic or obturator nerve endometriosis has been incorporated into the abstract and introduction to provide a comprehensive overview.
The manuscript has undergone thorough English language editing to enhance clarity and readability. Section 4.1 now includes the significance of "deep endometriosis," offering a rationale for its detailed exploration. Section 4 has been streamlined to focus on elements of higher novelty, as suggested. Section 5 has been restructured to separate diagnosis and treatment considerations, with neuropsychiatric aspects expanded for greater emphasis. Additionally, Sections 7, 8, and 9 have been integrated into Section 6 to create a cohesive discussion of deep infiltrating endometriosis (DIE). I am confident these changes address the concerns raised and enhance the overall quality of the manuscript. Thank you once again for your insights and guidance during this process.
Yours sincerely
Jerzy Leszek
Reviewer 2 Report
Comments and Suggestions for Authors
1. Again, the paper focuses on facts about endometriosis and, to a lesser extent, the association of endometriosis with neuropsychiatric comorbidities, as announced in the title. I suggest changing the title in a more general way about the endometriosis.
2. Table 1. Two sentences: " Both ROS and fibrosis play a significant role in modulating endometriosis lesions. However, there is still a necessity of deepening those topics to fully understand DE pathophysiology" can not be the title of the table.
For example, the title would be: Factors of Endometriosis Modulation (or something similar).
3. Tabel 2. Similar to objection 2, the title of Table 2 is not appropriate.
Author Response
Dear Reviewer,
thank You very much for your time spent on reading this work.
The title of the paper has been adjusted to reflect a broader focus on endometriosis while still addressing its neuropsychiatric comorbidities.
The title of Table 1 has been revised to: "Factors Influencing Endometriosis Modulation" to align with its content.
The title of Table 2 has been updated to: "Summary of Neuropsychiatric and Peripheral Nerve Associations with Endometriosis" to better represent its focus.
We are open to anny further modification.
Yours sincerely
Magdalena Pszczołowska
Round 3
Reviewer 1 Report
Comments and Suggestions for Authors
I have reviewed the changes you made in response to the feedback provided, and I am pleased to see that you address my comments thoroughly.
Reviewer 2 Report
Comments and Suggestions for Authors
With a new title and corrections, the paper is acceptable for publication.
Author Response
Dear Reviewer,
thank you very much.
Yours sincerely
Jerzy Leszek